# Points of View on the Tools for Genome/Gene Editing

**DOI:** 10.3390/ijms22189872

**Published:** 2021-09-13

**Authors:** Chin-Kai Chuang, Wei-Ming Lin

**Affiliations:** Animal Technology Research Center, Division of Animal Technology, Agricultural Technology Research Institute, No. 52, Kedong 2nd Rd., Zhunan Township, Miaoli County 35053, Taiwan; 1052016@mail.atri.org.tw

**Keywords:** CRISPR, Cas9, genome/gene-editing, extracellular vesicles

## Abstract

Theoretically, a DNA sequence-specific recognition protein that can distinguish a DNA sequence equal to or more than 16 bp could be unique to mammalian genomes. Long-sequence-specific nucleases, such as naturally occurring Homing Endonucleases and artificially engineered ZFN, TALEN, and Cas9-sgRNA, have been developed and widely applied in genome editing. In contrast to other counterparts, which recognize DNA target sites by the protein moieties themselves, Cas9 uses a single-guide RNA (sgRNA) as a template for DNA target recognition. Due to the simplicity in designing and synthesizing a sgRNA for a target site, Cas9-sgRNA has become the most current tool for genome editing. Moreover, the RNA-guided DNA recognition activity of Cas9-sgRNA is independent of both of the nuclease activities of it on the complementary strand by the HNH domain and the non-complementary strand by the RuvC domain, and HNH nuclease activity null mutant (H_840_A) and RuvC nuclease activity null mutant (D_10_A) were identified. In accompaniment with the sgRNA, Cas9, Cas9(D_10_A), Cas9(H_840_A), and Cas9(D_10_A, H_840_A) can be used to achieve double strand breakage, complementary strand breakage, non-complementary strand breakage, and no breakage on-target site, respectively. Based on such unique characteristics, many engineered enzyme activities, such as DNA methylation, histone methylation, histone acetylation, cytidine deamination, adenine deamination, and primer-directed mutation, could be introduced within or around the target site. In order to prevent off-targeting by the lasting expression of Cas9 derivatives, a lot of transient expression methods, including the direct delivery of Cas9-sgRNA riboprotein, were developed. The issue of biosafety is indispensable in in vivo applications; Cas9-sgRNA packaged into virus-like particles or extracellular vesicles have been designed and some in vivo therapeutic trials have been reported.

## 1. Preface

Besides the common gene knockout purpose, a major goal of genome/gene editing experiments is to precisely convert a selected DNA sequence into a new desired one in the native context of the whole genome. Before the advent of programmable nucleases, most of the cases of the site-specific changes by homology-directed repair (HDR) were executed under the single agent gene editing strategy utilizing single-stranded DNA oligonucleotide templates [1]. The efficiency of HDR could be highly improved when DNA cleavage occurred at or near the recombination site [2,3]. As a comparison of the ranges of genome sizes, those for amphibians and flowering plants are from 1E9 to 1E11 and from 1E8 to 1E11 bp, those of birds and mammals are quite narrow, being around 1E9 and 3E9 bp per ploidy, respectively [4,5]. If we want to precisely operate the genomes of livestock, such as cows, pigs, sheep, and chickens, a nuclease which is able to recognize DNA sequences of equal to or more than 16 bp, theoretically, is indispensable. (let 4*^n^* ≥ 3E9, then *n* ≥ 16). Natural or engineered site-specific nucleases suitable for these purposes and their applications on genome/gene editing are described in this paper.

## 2. Endonucleases for Genome Editing

### 2.1. Homing Endonuclease and Meganuclease

Since the original study in 1971 to determine the “ω” self-splicing intervening sequence, it was later recognized as a group I intron [6] located within the mitochondrial gene-encoding large ribosomal RNA of yeast, Saccharomyces cerevisiae [7]. Additionally, the inheritance of this intron was induced by a site-specific endonuclease (now termed I-SceI) encoded within the intron sequence [8]. Such nucleases which can recognize a unique DNA target site among the whole genome are now nomenclatured as Homing Endonucleases, defined as microbial DNA-cleaving enzymes that mobilize their own reading frames by generating double-strand breaks at specific genomic invasion sites. These proteins display an economy of size and yet recognize long DNA sequences (typically 20 to 30 base pairs) [9]. Based on the specific amino acid motif in the catalytic core, HEs were categorized into at least six families, with each being associated with a particular host range. HNH, GIY-YIG, and EDxHD families are largely constrained to phage hosts. PD-(D/E)xK, His-Cys Box, and LAGLIDADG families are encoded in bacterial, protistic, and archaeal/eukaryotic hosts, respectively [10,11,12]. The endonucleases of the LAGLIDADG HE family are often referred to as “meganucleases (MN)” [13]. The LAGLIDADG endonucleases exist both as homodimers (where the two identical protein subunits are each typically 160 to 200 residues in size with a αββαββα core fold in which a long anti-parallel pair of β strands are fitted into the major groove of recognition site), and as monomeric proteins, where a tandem repeat of two LAGLIDADG domains is connected by a variable peptide linker. Compared with their homodimeric cousins, the monomeric proteins are rather small; their individual domains are often only 100 to 120 residues in size and can recognize fully asymmetric DNA target sites. In addition to the benefits that the recognition site of MNs are quite long (18–24 bp) and highly specific and the sizes are small so as to be prepared and engineered [14,15] easily, the richest natural sources are another advantage. For the full spectrum of the mammalian genome, a bank of 3 billion MNs is theoretically needed to cover all possible recognition sites, or hundreds of thousands to overlay all genes. That is a tremendous task.

### 2.2. Zinc Finger Nuclease (ZFN)

In comparison to a typical type II restriction enzyme, whose recognition site is overlapped with a cutting site, in the type IIS, where “S” means shifted cleavage, the enzyme contains independent modules for a separated recognition site and cutting site, e.g., FokI contains a 382 a.a. N-terminal DNA recognition domain and a 196 a.a. C-terminal nuclease domain (FN) [16]. After the proof-of-concept work prepared chimeric restriction endonuclease by linking the site-specific DNA-binding homeodomain of Ubx with FN [17], the primitive concept of combining the site-specific DNA-binding zinc finger domains with FN, termed as ZFN hereafter, to be an editable hybrid restriction enzyme was launched by Dr. Srinivasan Chandrasegaran’s group at Johns Hopkins [18]. The zinc finger is a big superfamily of domains and C_2_H_2_ is the most common type of it. A β sheet-turn-β sheet-turn-α helix has a rigid structure, where two cysteine residues located within the first turn and two histidine residues at the C-terminus of the α helix are coordinated to chelate a zinc ion. The α helix is fitted into the major groove of the DNA double helix and the -1, 3, and 6 a.a. residues interact with three consecutive nucleotides of the sense strand in the 3′ to 5′ orientation, respectively [19]. The C_2_H_2_ zinc fingers can be recognized as independent trinucleotide binding modules and linked into a polypeptide to distinguish longer DNA binding. In theory, one can design a zinc finger for each of the 64 possible combinations of trinucleotides, and one can arrange such zinc fingers to compose a sequence-specific artificial protein for any segment of DNA [20]. Because the nuclease activity of FN is stringently present in a dimer form [21], a pair of ZFNs with recognition sites in a tail-to-tail orientation was demonstrated necessary to perform effective double-strand cutting activity, which was essential to enhance the probability of homologous recombination a thousand-fold in vivo [22]. The D_483_ and R_487_ residues of FokI were involved in the FN dimer formation by interacting with each other between the two subunits. FN carried a D_483_R mutation which led both of the 483 and 487 residues of FokI to be positively charged, termed as FN_RR_. On the other hand, FN carried a R_487_D mutation, which led both the 483 and 487 residues of FokI to be negatively charged, termed as FN_DD_. Unlike the wild-type FN, FN_RR_ and FN_DD_ cannot form homodimers themselves; however, they can form heterodimers efficiently [23]. Such phenomena were utilized to improve the cutting specificity by using a pair of complementary FNs for the up-stream and down-stream recognition arm, respectively, as well as to reduce the toxicity caused by the off-targeting by unwanted homo-dimer [24]. Although ZFN seems a powerful tool for genome editions, some drawbacks should be noted. Firstly, there are still no perfect matches between zinc finger proteins and DNA triplexes. The specificity between GC-rich triplexes and zinc fingers was calculated to be 73%, whereas it was only 50% for AT-rich triplexes and their zinc finger partners [25]. The DNA-binding specificity of a C_2_H_2_ zinc finger was also revealed to be influenced by neighboring ones [26].

### 2.3. Transcription Activator-Like Effector Nuclease (TALEN)

Since the isolation of the avrBs3 gene in the bacterial plant pathogen *Xanthomonas campestris* pv. *Versicatoria* [27], AvrBs3 protein was found to be injected into plant cells via a type III secretion system [28,29] and to act as a site-specific transcription activator-like effector (TALE) to reprogram host cells. The AvrBs3 protein is composed of 1163 a.a. with a translocation domain at the most N-terminal end of a 287 a.a. N-terminal region, a middle part which contains 17 units of 34 a.a. complete repeats following with a 20 a.a. half repeat, and a 278 a.a. C-terminal region containing nuclear localization signals (NLSs) and an acidic transcriptional activation domain (AD) [30]. The sequences of the 34 a.a. repeats performing an α helix-random coil-α helix structure are nearly identical, except the polymorphic 12th and 13th residues, which are known as the repeat variable di-residue (RVD) and specifically specify a single binding site nucleotide through direct interactions [31,32,33,34]. The specificity and affinity for each RVD to a nucleotide were systemically studied [35,36]. Theoretically, merely four kinds of repeat, each with RVDs recognizing G, A, T, and C, respectively, are necessary to construct a DNA-binding domain specific to any given sequence. The N-terminus 152 a.a. and C-terminus 215 a.a. of AvrBs3 protein could be removed to leave a core region with intrinsic DNA-binding activity, and this core region was used as a fundamental framework for TAL effector nuclease (TALEN) designation [37,38,39,40]. X-ray crystal data revealed that the amino acid residues 162 to 288 of AvrBs4 perform four cryptic repeats of helical bundles to interact with a nucleotide T on the sense strand of a DNA double helix immediately in front of the nucleotides recognized by the canonical repeats, and this region provides the majority of the energy required for high-affinity target binding [41].The RVDs of complete repeat modules bind consecutive nucleotides of sense strands in the 5′ to 3′ direction. The first residue in each RVD (the 12th of the repeat) orients away from DNA to interact with the backbone of the eighth residue of the repeat to stabilize the interhelical loop and allow the second residue of the RVD to project into the major groove of the DNA and make sequence-specific contact with a single nucleotide of the sense strand [42]. The most common RVDs are HD, NG, NN, and NI for C, T, G > A, and A, respectively, in which NN can be replaced by NH for NH is more specific to G but has less affinity [34,35,36,43]. Online tools for custom TALEs and TALENs, such as TALE-NT 2.0, designation and modular assembly methods relying on Golden Gate cloning have been developed, enabling researchers to make constructs in a few days [44,45,46,47,48]. Besides the FN nuclease domain, transcription regulatory domains and DNA modification enzymes can be engineered to the C-terminus of the sequence-specific TALE-core structure to create artificial gene regulatory factors [49,50].

### 2.4. CRISPR/Cas Nucleases

It was identified that the spacer sequences between identical repeats of the clustered regularly interspaced short palindromic repeat (CRISPR) loci of bacterial genomes might originate from plasmid and phage [51,52]. The CRISPR RNA and CRISPR-associated protein (Cas) systems are now confessed as key components governing bacterial adaptive immune response which consists of three main stages: adaptation, expression, and interference. When a bacterium was attacked by an invader, a short DNA fragment, termed a protospacer, which is neighbored by a protospacer-adjacent motif (PAM) of the invader, was processed by adaptation Cas members, such as Cas1 and Cas2, to be inserted into the 5′ end of a spacer-repeat CRISPR array embedded in the host genome as stored memory. Memory was retrieved as the CRISPR array was transcribed to produce a long precursor CRISPR RNA (pre-crRNA), which was processed by an expression factor, such Cas6 or RNase III, within the repeat region to create mature crRNA, which was incorporated with Cas effectors, such as Cas5, Cas7, Cas8, and Cas11, to yield an RNA-guided sequence-specific endonuclease in the interference stage [53,54]. According to the number of Cas protein subunits included in the effector endonuclease complex, the CRISPR-Cas systems belong to two classes, with multi-subunit effector complexes in class 1, which can be further divided into three types: type I, type III and type IV, and single-protein effectors in class 2, including type II, type V and type VI [55,56,57]. Besides crRNA and Cas9 protein, a trans-activating CRISPR RNA (tracrRNA) whose 5′ region is complemented with the repeat sequence of crRNA is critical to perform endonuclease activity in the type II CRISPR systems. The crRNA and tracrRNA could be engineered into one single-guided RNA (sgRNA) in accompaniment with Cas9 to restore full and specific endonuclease activity [58]. The best-characterized and applied Cas9 enzyme was originally isolated from *Steptococcus pyrogenes*, and was referred to as SpCas9, or even simply as Cas9. SpCas9 is a large 1368 a.a. multidomain protein with two distinct lobes: the recognition (REC) lobe and the nuclease (NUC) lobe, connected through an arginine-rich bridge helix (residue 56 to 93) and a disordered loop (residue 712 to 717). The REC lobe is composed of three α-helical domains (Hel-I, Hel-II, and Hel-III) and the NUC lobe contains HNH and RuvC-like nuclease domains, as well as a PAM-interacting (PI) C-terminal domain [59,60] (Figure 1A,B). The apo-Cas9 protein should be assembled with guide RNA (native crRNA-tracrRNA hybrid or sgRNA) to achieve site-specific DNA recognition and cleavage activities. The 20 nt spacer sequence of crRNA provided DNA target specificity and the tracrRNA conferred a crucial role in Cas9 protein recruitment. Once the PAM (NGG for SpCas9) directly adjacent to a protospacer target site was trapped by R1333 and R1335 of the Cas9-guide RNA complex, it triggered local DNA melting at the PAM-adjacent site. The PAM-proximal 10–12 nucleotides (nt), 3′-end of the 20 nt spacer sequence is absolutely critical for site specificity, and was referred to as seed region. The DNA cleavage activity of CRISPR-Cas9 was excited by the conformational change induced by the R-loop formation between target DNA and spacer RNA [61,62]. The target DNA strand complementary to spacer RNA was cut by the HNH nuclease domain and the non-target DNA strand by the RuvC nuclease domain to produce a blunt-ended double-strand breakage at 3 bp upstream to PAM [63,64]. Either D_10_A [58] or H_983_A [65] mutation destroyed the RuvC nuclease activity. On the other hand, D_839_A [66], H_840_A [58], and N_863_A [67] mutations could eliminate the HNH nuclease activity. These mutations did not influence the target site binding affinity of Cas9-sgRNA. Cas9 carrying the D_10_A mutation and D_10_A/H_840_A double mutations were termed nickase (nCas9) and dead enzyme (dCas9), respectively (Table 1). The dCas9 could be taken as a guide RNA-derived sequence-specific DNA-binding protein, like TALE described above, and coupled with DNA manipulation enzymes or transcriptional activating/inhibitory domains to be harnessed for various applications [64]. The amino acid residues interacting with the PAM bases could be engineered to generate new PAM so as to broaden the spectrum of target sites. Based on the structure-guided rational design, the wild-type D1135, R1335, and T1337 were converted to E, Q, and R, respectively; the PAM was shifted from NGG to NGA. Additionally, as D1135, G1218, R1335, and T1337 were converted to V, R, E, and R, respectively; the PAM became NGC [68]. An engineered SpCas9 bearing D_1135_L/S_1136_W/G_1218_Q/E_1219_Q/R_1335_Q/T_1337_R substitutions in PI domain (SpG) targeted NGN PAM. SpG was further engineered to carry A_61_R/L_1111_R/N_1317_R/A_1322_R/R_1333_P substitutions to near-PAMless (NRN > NYN) variants, termed SpRY, with full endonuclease activities [69] (Table 1).

Besides Cas9, which recognized G-rich PAM at 3′ end of protospacer, class 2 type V Cas12a (originally called Cpf1) effector enzymes also became attractive [73]. The long pre-crRNA was bound and processed by an intrinsic RNase activity of Cas12a protein to mature crRNA, which was composed of a repeat sequence at the 5′ end and spacer at the 3′ end. This characteristic was utilized to design multiple crRNA in a single RNA transcript [73,74]. A canonical TTTV PAM was at the 5′ end of a 23 bp protospacer. Only a short 42–44 nt crRNA, which was composed of 19 nt repeat and 23–25 nt spacers, was necessary to guide the Cas12a’s RNA-dependent endonuclease activity, of which DNA was cut at the PAM-distal end to leave 5′ protruding staggered ends. Like Cas9, the RuvC nuclease domain was involved in non-complementary strand cleavage, while a new Nuc domain, instead of the HNH domain, was used in Cas12a for complementary strand cleavage [75]. The size of *Lachnospiraceae bacterium* MA2020 Cas12a (LbCas12a) was merely 1206 a.a. and as active as the most widely used Cas12a isolated from *Acidaminococcus* sp. (AsCas12a, 1307 a.a.). Engineered LbCas12a with Q_571_K and C_1003_Y mutations, referred to as Lb2Cas12a, was more active and could recognize both TTTV and CTTV PAM motives [76].

Unlike ZFN and TALEN, which utilize interactions between amino acid residues and nucleotides for DNA target sites recognition, CRISPR-Cas effectors use absolutely specific nucleotides–nucleotides base pairing. Besides PAM and the seed region, where completely precise Watson–Crick base pairing is necessary between guide RNA and the DNA target site, no more than three mispairings are permitted in the non-seed region of the spacer. Although off-targeting concerns became less criticized, engineered Cas9 variants with higher DNA target specificity are still an ongoing issue. Reagrding the structure-guided aspect, positively charged residues in the non-target strand binding groove were changed to alanine to interrupt their interactions with the negatively charged deoxynucleic acid backbone. The combination of K_848_A, K_1003_A, and R_1060_A mutations were chosen to obtain eSpCas9(1.1), which displayed efficient and precise genome editing in human cells in a series of two successive nucleotide mismatches of a 20 nt guide by replacing the wild-type SpCas9 with the eSpCas9(1.1) [70] (Table 1). Another approach aimed to decrease the affinity between the Cas9-sgRNA complex and target DNA by changing the polar residues contacting the phosphate backbone of the target DNA strand to alanine. On a variant carrying N_497_A, R_661_A, Q_695_A, and Q_926_A, SpCas9-HF1, substitutions were performed with high on-target activity and reduced off-target editing. The times of off-targeting detected on the VEGFA site 2 and 3 were reduced from 144 to 21 and from 32 to 1, respectively, by the HF1 counterpart [71,72] (Table 1). On the other hand, the spacer of the guide RNA was engineered to reduce the off-target activity. In truncated sgRNA, Tru-Greene, utilizing a spacer sequence shorter than 20 nt complementary to a protospacer target could reduce the off-target efficiency for up to three orders and maintain a high level of on-target editing [77]. Adding a short hairpin (hp) RNA on the 5′ end of the spacer was an alternative method for increasing editing specificity. The 5′ end of spacer involved in hp formation had a stronger influence on the mismatched base pairing in off-target sites than in the matched on-target site [78].

## 3. Precise Genome Editing

### 3.1. With Double-Strand Breaks

Precise and efficient genome editing is urgently required in many fields, such as modern breeding and genetic disease therapy [79]. An important step in genome editing was achieved when a site-specific endonuclease against the genomic site of interest could be manufactured. However, the generation of DNA breakage did not directly lead to DNA editing, which took place in the processes of cellular responses to the breakage. The DNA breaks created by such endonuclease could be processed by the DNA repair system of the host cell in two pathways. Most frequently, the breaks were joined by an error-prone non-homologous end-joining (NHEJ) pathway in which modification enzymes, such as exonuclease, deoxynucleotide terminal transferase, single-stranded DNA specific endonuclease, DNA polymerase I, and DNA ligase, as well as DNA-binding factor Ku70/80 heterodimer, were recruited [80,81]. A few template-independent or random base pairs could have been deleted or inserted at the break (indel mutations). On the other hand, a donor template DNA was necessary for the less frequently occurring homology-directed repair (HDR) pathway, which was largely restricted to the G2 and S phases of the cell cycle [82]. A long 3′ protruding single strand was produced at first from the broken end and it would hybrid with the donor template DNA to form an extending D-loop. In addition to exonuclease and DNA polymerase, single-strand DNA-binding proteins and DNA translocase were involved in this process. The extending single strand crossed back to the other end of the break so as to repair the break with donor template sequence [83,84]. Many Cas9-based strategies were reported to enhance the probability of HDR, such as using a pair of nickases to create long 3′ overhang ends [85].

The error-prone NHEJ pathway is not restricted to dividing cells and it has been conveniently and effectively applied to gene targeting experiments in accompaniment with programmable endonucleases. The HDR is restricted to the G2 and S phases of dividing cells; therefore, gene knock-in methods utilizing the characteristics of NHEJ and programmable endonucleases were developed for quiescent cells. The precise integration into target chromosome (PITCh) system was assisted by microhomology-mediated-end-joining (MMEJ), which harnessed independent machinery from homologous recombination (HR) and required an extremely short homologous sequence of 5–25 bp for DSB repair, resulting in precise gene knock-in [86,87]. Another robust NHEJ-based method, homology-independent targeted integration (HITI), used one programmable endonuclease site on target DNA and two sites flanking the inserted DNA fragment in order to remove the insertion of unwanted orientation [88,89].

### 3.2. Without Double-Strand Breaks

For the aspect of safety, DNA editing methods without double-strand breaks (DSBs) and additional donor template DNA were developed. The characteristic R-loop structure of target DNA-guided RNA created in complex with Cas9 was applied to establish the DNA Base Editor systems. Since the sense (or non-target) strand of protospacer DNA was cut by the RuvC domain and the non-sense strand (or the target strand hybrid with guide RNA) was cut by the HNH domain, a single-stranded DNA specific cytidine deaminase, such as apolipoprotein B mRNA editing enzyme, catalytic polypeptide-like 1 (APOBEC1), coupled to the N-terminus of nCas9 which carried a D_10_A (RuvC-) mutation, could facilitate the deamination of the cytidines to uracils, which were taken as thymidines in DNA on the sense strand of a protospacer in a window of 3 to 6 nt, depending on the length of the linker between these two proteins. For example, a 16 a.a. XTEN linker [90] offered an efficient deamination window of 5 nt, corresponding to positions 4 to 8 of the protospacer (the positions were numbered as 1 to 20 for the protospacer and 21 to 23 for PAM). The uracil bases were removed by uracil DNA glycosylase (UDG) by the cellular DNA repairing system to avoid unwanted error-prone substitutions or indels [91]. Therefore, an 83 a.a. strong UDG inhibitor (UGI) from the bacterial phage PBS1 of *Bacillus subtilis* was fused to the C-terminus of the previous APOBEC1-XTEN-nCas9(D_10_A) through a 4 a.a. linker to obtain APOBEC1-XTEN-Cas9(D_10_A)-UGI, the third generation of cytidine base editor (CBD3). The nick created by the HNH domain enhanced the conversion of the corresponding G on the non-sense strand to A with the U on the sense strand as the template by the cellular DNA repairing system [92]. Other single-stranded DNA-specific cytidine deaminases, AID and CDA1, were found to be more effective to converse the C directly following G to U and two successive copies of linker-UGI provided better specificity [93]. A Cpf1 BE system was established as well. Dead LbCpf1(D_832_A/E_1106_A/D_1125_A) (dCpf1), the 16 a.a. XTEN linker, SGGS linker (l_4_), engineered rat APOBEC1(W_90_Y/R_126_E)(ApobecYE), SV40 NLS (NLS), and UGI were used to construct dCpf1-BE0: Apobec1-XTEN-dCpf1-NLS-l_4_-UGI-NLS and dCpf1-BE-YE: NLS-ApobecYE-XTEN-dCpf1-NLS-l_4_-UGI-NLS with high editing efficiencies and specificities [94] (Figure 1C).

Besides the CBDs, the adenine base editors (ABDs) were also developed. Because no natural enzyme with adenosine deaminase activity on ssDNA was known, Gaudelli et al. used a 167 a.a. *E. coli* tRNA adenine deaminase, TadA1, which was able to convert adenine to inosine (I) in the single-stranded anti-codon loop of tRNA^Arg^ as starting material to evolve a DNA adenine deaminase. TadA, which had been fused to the N-terminus of a dead Cas9(D_10_A/H_840_A) with a crRNA directing a Cam^R^(H_193_Y) mutation region, was artificially evolved in *E. coli* carrying this inactivated antibiotic-resistant gene. TadA mutants in the colonies that survived after antibiotic selection were isolated. After seven rounds of selection, a TadA1 mutant, termed as TadA-7.10 or TadA*, which carried W_23_R, H_36_L, P_48_A, R_51_L, L_84_F, A_106_V, D_108_N, H_123_Y, S_146_C, D_147_V, R_152_P, E_155_V, I_156_F, and K_157_N mutations with deoxyadenosine deaminase activity, was obtained. A bpNLS-TadA1-TadA*-nCas9(D_10_A)-bpNLS adenine base editor (ABE7.10) was demonstrated to work in mammalian cells [95,96]. The TadA-7.10 was further evolved to broaden its applicability. A bacterial phage M13 system, in which the essential capsid pIII gene expression was regulated under T7 promoter, and a T7 RNAP that carried R_57_* and Q_58_* stop codon mutations was installed in a helper plasmid. The TadA-7.10-dCas9 was taken as the initiator of evolution, with a greater ability to reverse the stop codon mutations back to normal ones, indicated by the infectious M13 phage package. A much more active deoxyadenosine deaminase clone, TadA-8e, carrying further A_109_S, T_111_R, D_119_N, H_122_N, Y_147_D, F_149_Y, T_166_I, and D_167_N substitutions, was sequenced. In contrast to ABE7.10, a wild TadA1 was not necessary to help TadA-8e folding in mammalian cells. The DNA deamination activity of bpNLS-TadA-8e-nCas9(D_10_A)-bpNLS adenine base editor (ABE8e) was about 590-fold higher than that of ABE7.10 [97,98]. Efforts to develop base editors for the simultaneous catalyzation of both cytosine and adenine base conversion, ACBEs, were reported recently [99,100,101].

Since the CBEs and ABEs could only resolve the C to T (G to A) and A to G (T to C) conversions, more versatile DNA editing tools are still required for all twelve transition and transversion mutations. The initial prototype prime editor (PE) included an M-MLV reverse transcriptase fused to the 3′ end of Cas9(H_840_A) nickase through a flexible linker, nCas9(H_840_A)-RT, as well as a prime editing guide RNA, pegRNA, a template RNA fused to the 3′ end of sgRNA (Figure 1D). The sense (or target) strand of protospacer was cleaved by RuvC domain between positions 17 and 18 to leave a 17 nt 3′ overhang ssDNA, which could be hybridized with the template part of the pegRNA as a primer and extended for 7–22 nt by the fused RT. To optimize the transversion efficiency, the length of the primer binding sequence was set to about 13 nt, with a template region for an extension of 10–20 nt, and three substitution mutations, D_200_N, L_603_W, and T_330_P, were introduced into RT. The addition of another Cas9(H_840_A) nickase with a nicking sgRNA (ngRNA) made a nick at the non-edited strand of the previous nCas9/pegRNA and recognizing an edited sequence in the second protospacer could increase the conversion specificities [102,103]. A user-friendly open software for pegRNA and ngRNA, PrimeDesign, is now available online [104].

## 4. Forms and Delivery Methods of Programmable Site-Specific Endonucleases/DNA Editors

The use of Genetically Modified Organisms (GMOs) is a seriously criticized issue in animal, botany, and environmental fields; therefore, genome/gene editing techniques without the risk of exogenous DNA integration were pursued. Gene delivery vehicles that would integrate into the host genome, such as retro/lenti-viral and transposon vectors, were prevented as possible. The forms of programmable endonucleases and the methods (routes) to deliver them into cells provide different performances. The programmable endonucleases, for example, Cas9 hereafter, or gene editors can be delivered in the forms of DNA, mRNA, and protein (RNP complex). Plasmid DNA can be prepared easily and cost-effectively. A time lag before the onset of an edition is unpreventable for transcription and translation when Cas9 is expressed in the format of plasmid DNA after transfection; however, as DNA is quite stable, the expression can be sustained for several days. However, the prolonged expression of CRISPR/Cas9 might increase the risk of off-target effects. In vitro transcribed Cas9 mRNA and sgRNA have a narrower window of expression because of the instability of RNA after delivery. Such characteristics are preferred to be applied in mammal breeding, since embryos of the 1-cell stage without de novo RNA transcription [105] were usually operated by microinjection. The delivered Cas9 mRNA and sgRNA were translated and assembled in the cytosol, enabling faster onset of gene editing than the plasmid form. Like the RNA form, the protein format of Cas9 could prevent the risk of foreign gene integration and it could achieve immediate and transient gene editing after delivery. The recombinant Cas9 protein produced by *E. coli* and Cas9/sgRNA RNP could be bulkily delivered into cells physically by conventional electroporation in vitro [106]. Some devices designed for single-cell electroporation with high efficiency and cell viability under low voltage have been developed [107]. In the nanofountain probe system, a cell was attached to a horizontal flat electrode. A hollow pyramidal electrode, with an aperture of 500 nm in diameter, which was filled with electrolyte and macro-molecules to be delivered, was controlled by a closed-loop micromanipulator to downwardly approach the upper cell membrane. Then, a short electric pulse was applied across the cell membrane to create a nanopore to let the macro-molecules transfer into the cell before it resealed [108,109,110]. Another case, the nanostraw system, which was built on vertical alumina nanostraws extending from a horizontal track-etched cell culture membrane, formed an array of hollow nanowires connected to an underlying microfluidic channel. Cellular engulfment of the nanostraws provided an intimate contact, significantly reducing the necessary electroporation voltage and increasing homogeneity over a large area [111]. The diameter of the nanostraws was 150 nm, the height by the amount of membrane etched was 1.5 to 2.5 μm and the surface density was 1 × 10^8^/cm^2^ to provide sufficient nanostraws per cell [112]. In the third example, the nanopore system, a track-etched polycarbonate filter membrane with nanopores of 100 nm in diameter on top of the delivery reagent was used to let cells settle on it. The local electric field was enhanced through the nanopores, and close contact between the cell membrane and the nanopores was determinant for efficient localized electroporation. Surface coatings for the polycarbonate filter membrane facilitated the tight cell membrane/nanopore contact [113].

## 5. Strategies for In Vivo Genome Editing

### 5.1. Delivery of Cas9/sgRNA RNP with Virus-Like Particles

The strategies to deliver the Cas9/sgRNA ribonucleoprotein (RNP) for in vivo genome editing are another challenge, as Gag polypeptide is the only retro/lenti-viral protein required for the assembly and release of the immature virus particles and the viral genome RNA is not necessary for capsid assembly and entry to host cells. An immature capsid consists of about 1500 to 3000 Gag precursor proteins, which are hydrolyzed by a protease encoded by the Pol gene to produce matrix protein (MA, p19), capsid protein (CA, p24), and nucleocapsid protein (NC, p7), as well as small peptides (SP1, SP2, and p6) [114,115]. The formation of virus-derived extracellular vesicles (EVs) are based on the self-assembly of the viral envelope protein, such as vesicular stomatitis virus glycoprotein (VSV-G), and the term virus-like particles (VLPs) is defined as both viral envelope proteins and viral structural proteins assembled together into EVs [116]. Cas9 protein could be incorporated into VLP by direct fusion with the Gag polyprotein, either at the 5′-end of HIV MA [117] or 3′-end of MLV NC [118] or p6 [119], with a peptidyl linker which was recognized by the viral protease encoded by the viral Pol gene. Plasmid vectors expressing Cas9-Gag/Pol accompanied by vectors expressing VSV-G and sgRNA were cotransfected into 293T cells to prepare Cas9/sgRNA RNP packaged in VLP. In the cases of Gag-Cas9 fusion, one more Gag/Pol expression vector was needed to provide viral protease required for the process of maturation. The Cas9/sgRNA RNP also could be incorporated into VLP indirectly by noncovalent interactions with an adaptor or bridge system. The interactions between aptamer, a sequence-specific oligonucleotide, and aptamer binding protein were utilized as adaptor systems. An RNA segment of the mom gene of bacteriophage Mu (com) was strongly and specifically associated with a phage Com protein (COM) [120,121]. The com sequence inserted into the tetraloop of sgRNA had little or no influence on the package and activity of the Cas9-sgRNA RNP complex and interacted with the COM protein fused to the Gag polyprotein in the 3′-part of NC [122,123,124]. Besides Gag polyprotein, the G3C, V153L, and G177E gene-modified NEF protein (1100 copies per capsid) [125,126] and VPR protein (550 copies per capsid) [127,128] could be used as a carrier for the aptamer binding proteins [129]. Rapamycin, which was found to bind the 12 kDa FK506 binding protein (FKBP12) with high affinity (dissociation constant, Kd = 0.2 to 0.4 nM) [130,131], and the FKBP12-rapamycin complex could strongly associate with the FKBP12-rapamycin binding (FRB) domain (a.a. 2025 to 2114) of FKBP12-rapamycin associated protein (FRAP, also known as mTOR) (dissociation constant, Kd = 4.2 to 12 nM) [130,132]. The FKBP12–rapamycin–FRB ternary complex [133] could be used as a bridge to link Gag and Cas9-sgRNA together, e.g., the recombinant protein of FKBP12 fused to the 5′-end of Gag could help the incorporation of the engineered Cas9-sgRNA RNP complex where the 5′-end of Cas9 was fused with an FRB_(T2098L)_ domain, which specifically bound the rapamycin analog AP21967 with high affinity into the capsid of a VLP in the presence of AP21967 [134] (Figure 1E).

### 5.2. Delivery of Cas9/sgRNA RNP with Extracellular Vesicles

Extracellular vesicles (EVs), which are secreted by nearly all types of healthy cells, were initially described as releasing unneeded compounds from cells, whilst now they are recognized as signaling vesicles that exchange components between cells [135,136]. Based on the routes of biogenesis, EVs can be separated into two main categories: exosomes and microvesicles. The exosomes are known as membrane vesicles (50 to 150 nm in diameter) which are generated in the endosomal system. The membrane of the early endosome, with the aid of endosomal sorting complex required for transport (ESCRT) machinery, begins to invaginate to generate intraluminal vesicles (ILVs), whose numbers are accumulated during endosome maturation. The late endosomes, also known as multivesicular bodies (MVBs), could be transported and fused with the plasma membrane to let the ILVs release to extracellular space as exosomes do [137]. CD63, as well as CD9, CD81, and Syntenin-1, were found to be highly enriched in exosomes [138]. Taking advantage of tetraspanin member CD63, which has four hydrophobic transmembrane domains and short intracellular N- and C-termini [139], CD63 was engineered to carry each COM aptamer binding domain at both N- and C-ends in order to trap the Cas9/sgRNA complex, where the tetraloop of sgRNA was replaced by the com aptamer RNA into exosomes [140] (Figure 1F). CD63-GFP also could enhance the incorporation of the Cas9/sgRNA complex, where the C-terminus of Cas9 was fused with a nanobody against GFP into exosomes [141].

Microvesicles (MVs), also referred to as shedding vesicles, ectosomes, oncosomes, shedding bodies, and microparticles, on the other hand, are formed by direct budding from the plasma membrane. During the biogenesis of microvesicles, vertical trafficking of molecular cargoes to the plasma membrane, redistribution of membrane lipids, and the recruitment of the contractile machinery at the surface to allow for vesicle pinching are involved. The sorting of molecular cargo components is tightly regulated by small GTPases, such as ARF6 and Rab22a [142]. Ectopically expressed reporter proteins, mRNA and siRNA, are efficiently incorporated into microvesicles [143], as the stimuli that cause plasma membrane budding are heterogenously reflected by the wide distribution of the diameter of MVs from 50 to 1000+ nm [135]. A subtype of MVs, the arrestin domain-containing protein-1 (ARRDC1)-mediated microvesicles (ARMMs), with a more even diameter around 50 nm was engineered as a platform for the delivery of macromolecules [144].

## 6. Concluding Remarks and Perspectives

Since the publication of the seminal paper that showed that simply a fused sgRNA is enough for Cas9 to obtain full ability to recognize a target site, the Cas9-sgRNA system has become the most popular genome-editing tool. Tremendous efforts to broaden the versatilities of applications and to improve the recognition accuracy of Cas9 have been made. To decrease the off-target risk by prolonged expression, methods to transiently express Cas9-sgRNA RNP or to directly use it are being accumulated. Besides the biosafety and effectivity concerns, convenience and cost are other criteria to be overcome for the future novel concept of delivering Cas9-sgRNA for in vivo treatments. Vesicles to precisely deliver Cas9-sgRNA to selected tissues or even cells of interest are goals to be pursued.

## Figures and Tables

**Figure 1 ijms-22-09872-f001:**
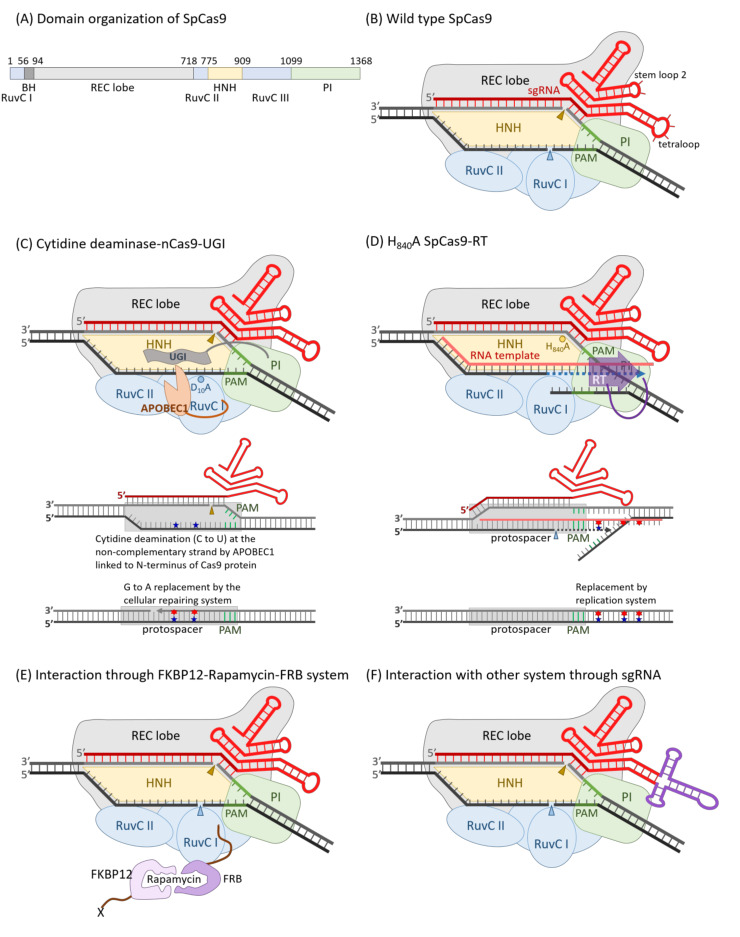
Diagrams of SpCas9 and its derivatives for various applications. The domain organization of SpCas9 (**A**) and a schematic diagram of wild-type SpCas9 associated with a sgRNA (**B**) was illustrated. The non-complementary strand is cut by the RuvC nuclease domain, and this nuclease activity was blocked in D_10_A mutant. On the other hand, the complementary strand was digested by the HNH nuclease domain, and such nuclease activity was destroyed in H_840_A mutant. (**C**) The D_10_A mutant, also named Cas9 nickase (nCas9), was engineered as a C to T nucleotide editor by linking a cytidine deaminase, APOBEC1, on the N-terminus of it and the switching probability could be elevated by the fusion of a uracil glycosylase inhibitor (UGI) on the C-terminus of nCas9. Like TALE, dCas9 could be guided by a sgRNA as a sequence-specific DNA-binding riboprotein. Transcriptional regulators, DNA modification enzymes, or histone modification enzymes could be fused to either or both of the N- and C-termini. In case of reverse transcriptase, it was fused to the C-terminus of Cas9, accompanied by an RNA template with 3′-end complementary to the non-complementary strand of protospacer, which could alter the nearby nucleotides downstream the RuvC cutting site (**D**). The localization of the Cas9-sgRNA also could be guided to an X-protein through an FKBP12–rapamycin–FRB bridge (**E**). The localization of the Cas9-sgRNA could also be guided via certain specific interactions, such as those between aptamer RNA and ABP. The tetraloop was replaced by an RNA aptamer of unique secondary structure, which can be recognized by a specific aptamer binding protein (**F**).

**Table 1 ijms-22-09872-t001:** The engineered mutations of SpCas9 on target site recognition and nuclease activities.

	Domain Where Mutations Engineered							Functional Changes	Reference
Mutants	RuvC-I (1–55)	BH (56–93)	REC Lobe (94–717)	RuvC-II (718–774)	HNH (775–908)	RuvC-III (909–1098)	PI (1099–1368)		
SpG							D_1135_L, S_1136_W, G_1218_Q, E_1219_Q, R_1335_Q, T_1337_R	The recognition sequence of PAM changed from NGG to NGN.	[69]
SpRY		A_61_R					L_1111_R, D_1135_L S_1136_W, G_1218_Q E_1219_Q, N_1317_R, A_1322_R, R_1333_P, R_1335_Q, T_1337_R	The recognition sequence of PAM changed from NGG to PAMless (NRN > NYN).	
eSpCas9(1.1)					K_848_A	K_1003_A R_1060_A		eSpCas9(1.1) displayed efficient and precise genome editing in human cells.	[70]
SpCas9-HF1			N_497_A R_661_A Q_695_A			Q_926_A		SpCas9-HF1 performed with high on-target activity and reduced off-target editing.	[71,72]
nSpCas9	D_10_A							RuvC nuclease activity was eliminated. (nickase)	[58]
SpCas9(H_840_A)					H_840_A			HNH nuclease activity was diminished.	
dSpCas9	D_10_A				H_840_A			Dead SpCas9 lost both of the RuvC and HNH nuclease activities.	
SpCas9(N_863_A)					N_863_A			HNH nuclease activity was eliminated.	[67]
SpCas9(D_839_A)					D_839_A			HNH nuclease activity was diminished.	[66]
SpCas9(H_983_A)						H_983_A		RuvC nuclease activity was eliminated.	[65]

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
