# Peer review of "Points of View on the Tools for Genome/Gene Editing"

_ijms, 2021, doi:10.3390/ijms22189872_

Round 1

Reviewer 1 Report

Chuang and Lin overviewed the genome/gene-editing technology.

The submitted manuscript is already well-categorized and adequately summarized.

I hope it to be published in this journal after minor revision.

Minor concerns:

line 33: What’s the meaning of “per bloody”?

line 225-232: In case of eSpCas9(1.1) and SpCas9-HF1, it would be better to show the quantitative data of enhancements in the efficiency and the off-targeting rate compared to SpCas9.

Table 1, N863A, ref [64]: Mutant name is missing.

Reviewer 2 Report

Overview: The manuscript by Chuang and Lin aims to provide a review of the gene editing technologies that are now being used as genetic tools to manipulate eukaryotic genomes.  The authors describe, in detail, the vast array of DNA nucleases, repurposed for use as gene editing tools.  The authors continue to provide extensive information regarding the evolution of editing tools that do not require double-stranded breaks.  A discussion surrounding delivery methods and strategies for in vivo genome editing including the use of the now popular RNP provide a sufficient end to the paper.  The figure is a bit complex but informative providing a unique opportunity for readers to compare iterations of gene editing tool side-by-side.  The authors appear to have expertise in protein design and cloning which does provide some significant weight to the quality of the discussion on programmable nucleases.

Critique: The strength of this manuscript is the detailed discussion surrounding endonucleases, including an informative history beginning with the infamous, Meganuclease.  I commend the authors for their lucid description of each of the double-strand break and base editing tools.  I suggest that the authors change the title of this manuscript to reveal only a review of those enzymes and enzymatic activities, because the rest of the manuscript is poorly conceived and inaccurate.  Many important discoveries have been left out and the history of gene editing, specifically single agent gene editing, is misrepresented.  I suspect these authors are not familiar with the true origins of gene editing nor the work that was done prior to the discovery, transformation, and maturation of CRISPR/Cas (the genetic tool that had simply obliterated the others).  It is a common mistake among people who have come into the field late.  Let me provide you with a couple of important points should you decide to undertake a very major revision if you maintain the current theme of the manuscript.

  1. On the front page online 29, the statement… The goal of gene editing is to precisely convert…. This leads one to believe that the only form of gene editing is to alter a DNA sequence, yet the most important roles for the programmable enzymes now is genetic knockout, i.e., inducing the loss of a specific sequence or the engagement of a frame shift mutation. 
  2. Section 3.1 ; the use of double-stranded DNA breaks was not the first important step in DNA genome editing. Single-stranded oligonucleotide's have been used as single agents to convert point mutations to wild-type for about 25 years in the field actually predicted most of the metabolic pathways that we now see in the more famous CRISPR/Cas systems.  If the authors continue along this path, I certainly would include a long and detailed description at the  beginning of  the \Precise Genome Editing section that describes gene editing without double-strand breaks; and I do not mean based editing… There are number of prior nearing labs that I read routinely that did break through work on educating us about gene editing…… Kmiec, Olson. Krauss, Glazer, Wolff are among the few major contributors that come to mind; these   contributions not only set the stage for gene editing but also identified many reaction parameters that have been put in use even in human trials.
  3. At the end of that section line 273…. I have read some very interesting work from Sansbury and colleagues who developed a cell free extract system to study the simultaneously occurring NHEJ and HDR.  That work should be referenced there.
  4. A fascinating paper by Cruz-Beccera and Kadonaga ( eLife, 2020) is a wonderful description of how best to enhance HDR and it does not include the elevation of ds breaks. Add this work too.

These are about a few of the additions that need to be covered in this manuscript if the title would be truly representative of what is in the paper.  The field began by work by Fred Sherman and Joe Walder and if you want a historical treatment to be fair, those original experiments in yeast and in bacterial cells should be site.

Based on this review , I would recommend rejection at this time with a major revision… The major revision would include either the addition of the appropriate references and historical accuracy or elimination of all of that with a focus specifically on the protein part of gene editing.

Round 2

Reviewer 2 Report

.The manuscript has been sufficiently modified and is now acceptable for publication